# Structural basis of FPR2 in recognition of Aβ$_{42}$ and neuroprotection by humanin

Ya Zhu[1,7], Xiaowen Lin[1,2,3,7], Xin Zong[1,3,7], Shuo Han[1,2,3,7], Mu Wang[1,4], Yuxuan Su[5], Limin Ma[1], Xiaojing Chu[1], Cuiying Yi[1], Qiang Zhao [1,3,6✉] & Beili Wu [1,2,3,4✉]

Formyl peptide receptor 2 (FPR2) has been shown to mediate the cytotoxic effects of the β amyloid peptide Aβ$_{42}$ and serves as a receptor for humanin, a peptide that protects neuronal cells from damage by Aβ$_{42}$, implying its involvement in the pathogenesis of Alzheimer's disease (AD). However, the interaction pattern between FPR2 and Aβ$_{42}$ or humanin remains unknown. Here we report the structures of FPR2 bound to G$_i$ and Aβ$_{42}$ or *N*-formyl humanin (fHN). Combined with functional data, the structures reveal two critical regions that govern recognition and activity of Aβ$_{42}$ and fHN, including a polar binding cavity within the receptor helical bundle and a hydrophobic binding groove in the extracellular region. In addition, the structures of FPR2 and FPR1 in complex with different formyl peptides were determined, providing insights into ligand recognition and selectivity of the FPR family. These findings uncover key factors that define the functionality of FPR2 in AD and other inflammatory diseases and would enable drug development.

[1] CAS Key Laboratory of Receptor Research, State Key Laboratory of Drug Research, Shanghai Institute of Materia Medica, Chinese Academy of Sciences, Shanghai, China. [2] School of Pharmaceutical Science and Technology, Hangzhou Institute for Advanced Study, University of Chinese Academy of Sciences, Hangzhou, China. [3] University of Chinese Academy of Sciences, Beijing, China. [4] School of Life Science and Technology, ShanghaiTech University, Shanghai, China. [5] School of Basic Medicine and Clinical Pharmacy, China Pharmaceutical University, Nanjing, China. [6] Zhongshan Institute for Drug Discovery, Shanghai Institute of Materia Medica, CAS, Zhongshan, China. [7] These authors contributed equally: Ya Zhu, Xiaowen Lin, Xin Zong, Shuo Han. ✉email: zhaoq@simm.ac.cn; beiliwu@simm.ac.cn

In response to the *N*-formyl peptides from microbes and mitochondria, FPRs play crucial roles in host defense and inflammation[1]. Although FPR1 and FPR2 share 69% sequence identity, these two receptors display distinct ligand-binding preferences. FPR1 preferentially recognizes short *N*-formyl peptides such as the *E. coli*-derived peptide *N*-formyl–Met–Leu–Phe (fMLF), and is associated with antibacterial inflammation and metastasis of malignant glioma cells[2]. In contrast, FPR2 is promiscuous in ligand binding with recognition to a vast array of ligands with diverse sizes, structures and functions, ranging from small molecules and lipids to peptides and proteins, which enable its involvement in chronic inflammatory diseases such as AD, systemic amyloidosis, and atherosclerosis[2].

AD is the most common neurodegenerative disease and is characterized by overproduction of β-amyloid peptides (Aβ) in the brain[3]. Aβ$_{42}$, a major causative factor of AD, plays a central role in inducing neurotoxicity and formation of senile plaques[3]. A bulk of evidence suggested that FPR2 served as a receptor mediating the Aβ$_{42}$-elicited proinflammatory responses that have an important role in the pathogenic process of AD[4–6]. FPR2 has also been shown to promote internalization of Aβ$_{42}$ to facilitate its uptake and fibrillary aggregation in mononuclear phagocytes[5]. Humanin (HN), a 24-amino acid polypeptide, was discovered to protect neuronal cells from apoptosis induced by Aβ, with proven effects on cell survival, metabolism, response to stressors, and inflammation[7,8]. Further investigation identified FPR2 as a functional receptor of HN, providing a foundation for this peptide suppressing the effect of Aβ$_{42}$ by competitively binding to FPR2[9]. The involvement of FPR2 in both the Aβ$_{42}$ cytotoxicity and HN neuroprotection demonstrates its importance in the pathogenesis of AD and offers promise of this receptor as a potential drug target for AD. However, the molecular basis of FPR2 in mediating the actions of Aβ$_{42}$ and HN is unknown, which hampers understanding of its functionality in AD and discovery of drugs.

In this work, we determined the cryo-electron microscopy (cryo-EM) structures of FPR2 bound to G$_i$ and four peptide agonists with diverse sequences and lengths, including Aβ$_{42}$, fHN, and two formyl peptides, *N*-formyl–MLFII (fM5, derivative of bacteria-derived formyl peptide) and *N*-formyl–MYFINILTL (fM9, mitochondria-derived formyl peptide). The Aβ$_{42}$–FPR2–G$_{i2}$ structure for the first time provides molecular details that define the recognition of Aβ$_{42}$, the most toxic form of Aβ, with its receptor. Furthermore, we also solved the structure of FPR1 in complex with fMLF and G$_i$ to better elucidate the ligand-recognition modes of the FPR family.

## Results and discussion

**Conserved activation mode of FPRs by distinct peptide agonists.** To enable structure determination of FPR2, a thermostable construct was generated by introducing a mutation S211$^{5.48}$L (superscript indicates residue numbering using the Ballesteros–Weinstein nomenclature[10]) and truncating 5 C-terminal residues (E347–M351) (Supplementary Fig. 1a). To further elevate protein yield, a thermostable apocytochrome b$_{562}$RIL fusion protein[11] was connected to the receptor N terminus using the tobacco etch virus (TEV) protease cleavage site as a linker, which facilitated removal of the fusion protein during complex preparation (Supplementary Fig. 1a). For FPR1, the flexible C terminus of the receptor (R322–K350) was truncated to improve protein yield and homogeneity (Supplementary Fig. 1b). Ligand-binding data indicate that these modifications have little effect on recognition of the peptide agonists (Supplementary Table 1). To optimize protein homogeneity, G$_{i1}$ and G$_{i2}$ were used to form complexes with FPR1 and FPR2, respectively

(Supplementary Fig. 1c). The structures of FPR2 bound to G$_{i2}$ and distinct peptide agonists, Aβ$_{42}$, fHN, fM9, and fM5, and the fMLF–FPR1–G$_{i1}$ structure, were determined by cryo-EM single-particle analysis at 2.8–3.3 Å resolutions (Fig. 1a; Supplementary Figs. 1d-r, 2, 3; Supplementary Table 2).

Upon binding to the peptide agonists, FPR1 and FPR2 adopt an active conformation similar to that in the previously determined crystal structure of FPR2 bound to the highly potent agonist WKYMVm[12], with Cα r.m.s.d. of 0.9–1.4 Å (Supplementary Fig. 4a). Compared with the inactive structure of the C5a receptor (C5aR)[13], which shares the highest sequence similarity with the FPRs among the G-protein-coupled receptors (GPCRs) with known structures, the peptide–FPR–G$_i$ structures exhibit a 9-Å outward movement of helix VI and an inward shift of helix VII by approximately 6 Å, which enable G$_i$-protein coupling (Supplementary Fig. 4a). Alignment of the active FPR structures reveals an overlap of the backbones for both the receptor and G protein (Supplementary Fig. 4b), and the receptor–G$_i$ interactions are mostly conserved in these structures, suggesting a common activation mode of the two FPRs.

Albeit with diversity in both sequence and length, fM5, fM9, fHN, and Aβ$_{42}$ share a deep binding cavity shaped by the extracellular loops and helices II, III, V, VI, and VII in FPR2, with their N termini penetrating into the pocket within the receptor transmembrane helical bundle (Fig. 1b). The side chains of the N-terminal formylated methionine in fM5, fM9, and fHN and the residue D1 of Aβ$_{42}$ insert into a conserved binding crevice between helices III and VI at the bottom of the ligand-binding pocket, forming contacts with L109$^{3.36}$, F110$^{3.37}$, V113$^{3.40}$, W254$^{6.48}$, F257$^{6.51}$, and Q258$^{6.52}$ (Fig. 1c). A similar interaction pattern was also observed between FPR1 and the N terminus of the tripeptide fMLF (Fig. 1c). Compared with the inactive C5aR structure, the FPR structures reveal notable conformational differences in this region, including the rotamer conformational changes of the conserved class-A GPCR "toggle switch" W$^{6.48}$ and P$^{5.50}$–I/V$^{3.40}$–F$^{6.44}$ motif (Supplementary Fig. 4c). Given the fact that helices III and VI are largely involved in GPCR activation[14], this structural feature suggests that the interactions between the peptide N terminus and FPRs may play an important role in modulating receptor signaling. Indeed, alanine mutations in this region substantially decreased agonist potency of the peptide ligands at the FPRs in an inositol-phosphate (IP) accumulation assay (Fig. 1e–h; Supplementary Fig. 5a–d; Supplementary Table 3). These structural and functional data imply that the FPRs adopt a conserved activation mode when stimulated by distinct peptide agonists.

The N-terminal formyl group of the formyl peptides has been suggested to be essential for bioactivity of these chemotactic peptides[1]. In the structures of formyl peptide–FPR–G$_i$ complexes, the *N*-formyl group, together with the main chain of the residue fM1, forms a polar interaction network with three charged residues D106$^{3.33}$, R201$^{5.38}$, and R205$^{5.42}$ in FPR1 and FPR2 (Fig. 1d). The importance of this binding mode in mediating agonistic activity of the formyl peptides was reflected by a drastic reduction of agonist potency of fM9 and fHN at FPR2, as well as fMLF at FPR1 for the mutants D106$^{3.33}$A, R201$^{5.38}$A, and R205$^{5.42}$A (Fig. 1e–g; Supplementary Fig. 5e, f, h; Supplementary Table 3). In contrast, these three mutations displayed a limited effect on Aβ$_{42}$-induced FPR2 signaling (Fig. 1h; Supplementary Fig. 5g; Supplementary Table 3). This aligns well with the Aβ$_{42}$–FPR2–G$_{i2}$ structure, in which the polar network is absent due to lack of the *N*-formyl modification in Aβ$_{42}$ (Fig. 1d).

**The N terminus of Aβ$_{42}$ is key for FPR2 recognition.** Upon binding to FPR2, the N-terminal residues D1–Y10 of Aβ$_{42}$ exhibit

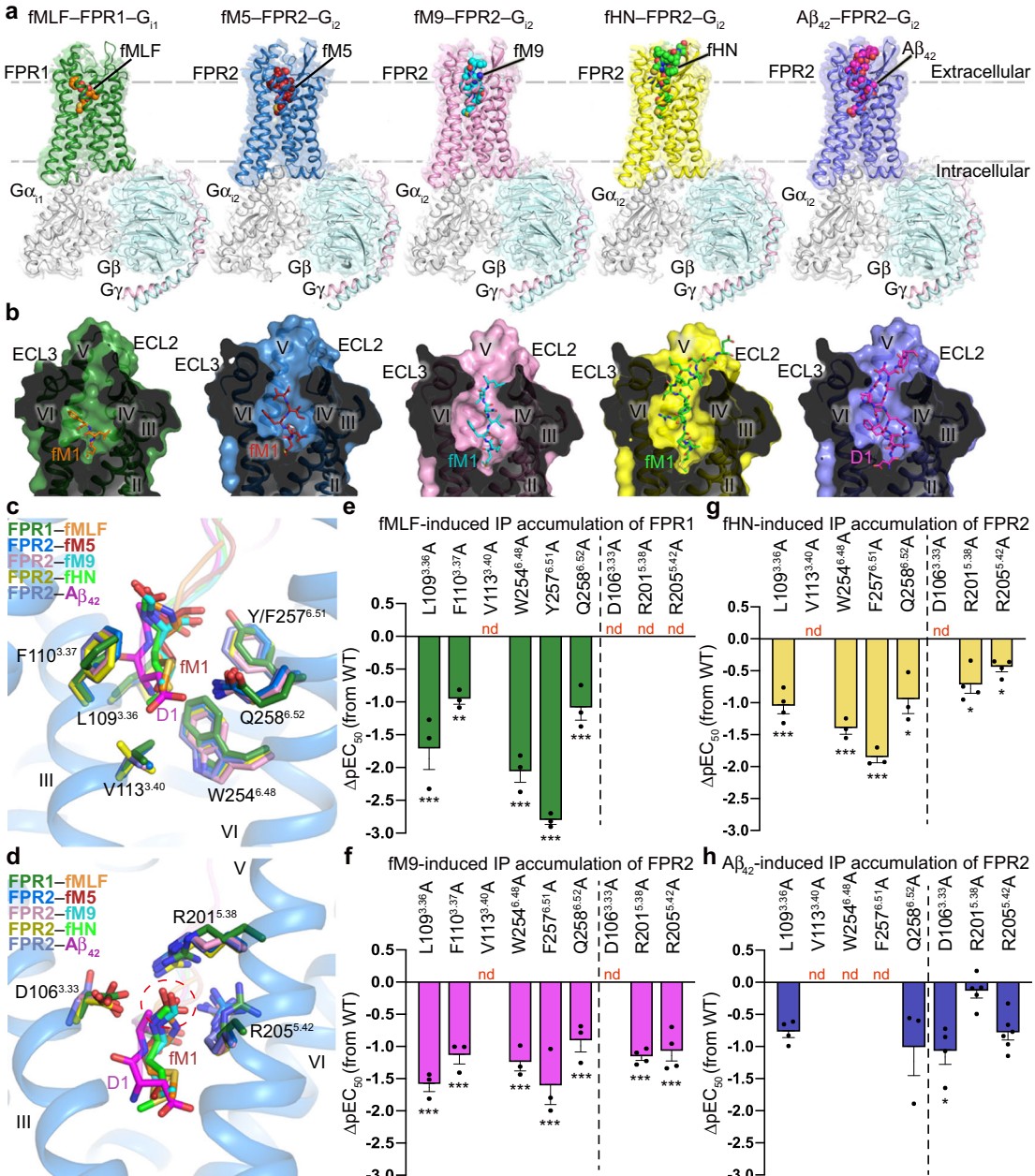

**Fig. 1 Overall structures and ligand-binding pockets in peptide agonist–FPR–$G_i$ complexes. a** Overall structures of fMLF–FPR1–$G_{i1}$, fM5–FPR2–$G_{i2}$, fM9–FPR2–$G_{i2}$, fHN–FPR2–$G_{i2}$, and $A\beta_{42}$–FPR2–$G_{i2}$ complexes. The cryo-EM maps and structures are colored according to chains. The peptide ligands are shown as spheres. **b** Cut-away view of ligand-binding pockets in the peptide agonist–FPR–$G_i$ structures. The receptors are shown as surface and cartoon representations. The ligands are shown as sticks. **c, d** Interactions between the FPRs and the N termini of the peptide agonists. The N-terminal residue fM1 of the peptides in the structures of fMLF–FPR1–$G_{i1}$, fM5–FPR2–$G_{i2}$, fM9–FPR2–$G_{i2}$, and fHN–FPR2–$G_{i2}$, and the N-terminal peptide residues D1 and A2 in the $A\beta_{42}$–FPR2–$G_{i2}$ structure are shown as sticks. The receptor residues that interact with the peptide N termini are also shown as sticks. Only the receptor in the fM5–FPR2–$G_{i2}$ structure is shown in blue cartoon representation for clarity. **c** Interactions between the FPRs and the side chains of the peptide N termini. **d** Interactions between the FPRs and the N-formyl groups at the N termini of the peptides. The N-formyl groups are highlighted by a red dashed circle. **e–h** Peptide agonist-induced IP accumulation of FPR1 and FPR2 mutants. Bars represent differences in calculated peptide agonist potency (p$EC_{50}$) for each mutant relative to the wild-type receptor (WT). Data are shown as mean ± SEM (bars) from at least three independent experiments performed in triplicate with individual data points shown (dots). *$P < 0.05$, **$P < 0.001$, ***$P < 0.0001$ by one-way analysis of variance followed by Dunnett's post-test compared with the response of the wild-type receptor. Supplementary Table 3 provides detailed statistical evaluation, $P$-values, numbers of independent experiments ($n$), and expression levels. Source data are provided as a Source Data file.

an elongated conformation and adopt a binding pose nearly perpendicular to the membrane plane, fitting into a binding groove bordered by helices III, V, VI, and VII, and the second and third extracellular loops (ECL2 and ECL3) (Fig. 2a). The cryo-EM map also reveals additional densities for six residues, which may

belong to the C terminus of $A\beta_{42}$. These residues form a β-sheet structure with the N-terminal segment of the peptide and make extra contacts with the extracellular loops of FPR2 (Fig. 2a and Supplementary Fig. 3e). The rest of the peptide was not traced due to the absence of densities in the cryo-EM map. Our ligand-

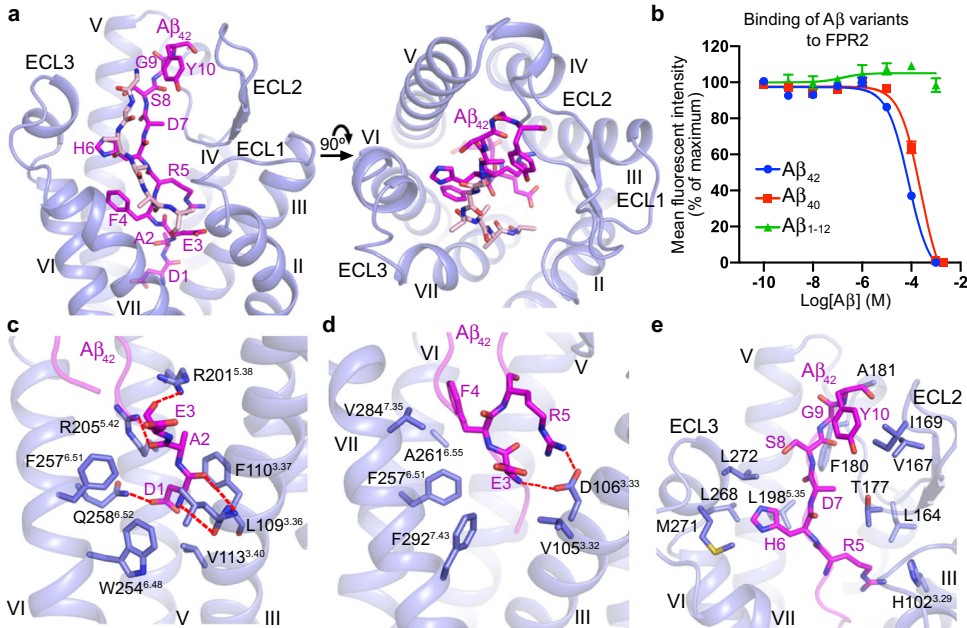

**Fig. 2 Recognition of Aβ₄₂ at FPR2. a** Binding pocket for Aβ₄₂ in FPR2. The Aβ₄₂–FPR2–G_{i2} structure is shown in both side (left) and extracellular (right) views. Aβ₄₂ is shown as sticks and colored magenta (N-terminal part) and pink (C-terminal part). **b** Inhibition of WK(FITC)YMVm binding to wild-type FPR2 by Aβ₄₂, Aβ₄₀, or Aβ₁₋₁₂. Data are displayed as mean ± SEM from at least three independent experiments (*n*) performed in triplicate (Aβ₄₂, *n* = 18; Aβ₄₀ and Aβ₁₋₁₂, *n* = 3). Source data are provided as a Source Data file. **c–e**, Interactions between FPR2 and Aβ₄₂. Aβ₄₂ residues and the receptor residues that are involved in interactions are shown as sticks. Polar interactions are shown as red dashed lines. **c** Interactions between FPR2 and the Aβ₄₂ residues D1–E3. **d** Interactions between FPR2 and the Aβ₄₂ residues E3–R5. **e** Interactions between FPR2 and the Aβ₄₂ residues R5–Y10.

binding assay showed that Aβ₄₂ and Aβ₄₀, another major form of Aβ with two residues shorter at the C terminus, bound to FPR2 with comparable binding affinities, while the N-terminal fragment Aβ₁₋₁₂ was incapable of binding to the receptor (Fig. 2b). In addition, Aβ₄₀ displays an only mild reduction of potency in inducing FPR2 signaling compared with Aβ₄₂ (3-fold reduction of EC₅₀; Supplementary Fig. 5i and Supplementary Table 3). These data suggest that the last two amino acids of Aβ₄₂ do not contribute much to its binding and agonist activity at FPR2, but the N-terminal region itself is not enough for recognizing the receptor. The other parts of Aβ₄₂ may play a role in stabilizing the conformation of the peptide N terminus to enable FPR2 binding and/or is key for initial receptor–peptide recognition. While previous structural studies of Aβ revealed multiple conformations of the peptides[15], the Aβ₄₂–FPR2–G_{i2} structure indicates that a specific form of Aβ₄₂ is required for binding to FPR2.

The N-terminal region of Aβ₄₂ acts as a major player in coupling to FPR2. The residues D1 and A2 squeeze into a narrow "channel" at the bottom of the ligand-binding pocket and play a critical role in triggering receptor activation as discussed above. This subpocket, which is formed by helices III, V, and VI, accommodates the peptide N terminus mainly through hydrophobic contacts (Fig. 2c). The only polar interactions are hydrogen bonds between side chains of the peptide residue D1 and the residue Q258^{6.52} in the receptor, as well as between main chains of D1 and the FPR2 residues L109^{3.36} and F110^{3.37} (Fig. 2c). This aligns with a 4-fold reduction of the Aβ₄₂ potency in inducing IP accumulation at the FPR2 mutant Q258^{6.52}A (Supplementary Fig. 5c and Supplementary Table 3). The polar residues R201^{5.38} and R205^{5.42} in FPR2, which are involved in a polar-interaction network with the N-terminal formylated methionine of the formyl peptides, form hydrogen-bond interactions with main chains of the Aβ₄₂ residues A2 and E3 due to a deeper binding position of Aβ₄₂ compared with the formyl peptides and the absence of *N*-formyl group in Aβ₄₂ (Figs. 1d, 2c).

Unlike the substantial impairment of fM9- or fHN-induced cell signaling, the FPR2 mutants R201^{5.38}A and R205^{5.42}A exhibit wild-type level of IP production when activated by Aβ₄₂ (Fig. 1h; Supplementary Fig. 5e–g; Supplementary Table 3), suggesting a less important role of these two basic residues in mediating the agonistic activity of Aβ₄₂. Alanine replacement of the residue D106^{3.33} displayed a 7-fold reduction of Aβ₄₂ potency, which may reflect the importance of a salt bridge between this acidic residue and the peptide residue R5 and a hydrogen bond with the main chain of E3 (Figs. 1h, 2d; Supplementary Fig. 5g; Supplementary Table 3). The bulky residue F4 in Aβ₄₂ packs into a shallow subpocket shaped by helices VI and VII, forming hydrophobic contacts with F257^{6.51}, A261^{6.55}, and V284^{7.35} (Fig. 2d). This binding mode is supported by a drastic impairment of the agonistic activity of Aβ₄₂ at the mutants F257^{6.51}A and V284^{7.35}A (Supplementary Fig. 5c, i; Supplementary Table 3). The importance of the N terminus of Aβ₄₂ in mediating its activity was further underlined by a substantially reduced potency for the Aβ₄₂ variants D1A and E3A (Supplementary Fig. 5i and Supplementary Table 3).

In addition to the transmembrane helical bundle, the extracellular loops of FPR2 largely contribute to the binding of Aβ₄₂. Residues R5–Y10 of the peptide extend into a narrow binding groove on the extracellular side of the receptor, forming extensive hydrophobic contacts with ECL2 and ECL3 (Fig. 2e). The binding interface is composed of two regions: one is formed by the receptor ECL2 and the Aβ₄₂ residues R5 and D7–Y10, while the other one includes ECL3 and the residue H6 in the peptide (Fig. 2e). I169W and F180A, mutations of two residues located at the entrance to the binding groove, resulted in a decreased agonist potency of Aβ₄₂ in the IP accumulation assay (Fig. 2e; Supplementary Fig. 5i; Supplementary Table 3). These data demonstrate the importance of the extracellular region of FPR2 in governing the agonistic activity of Aβ₄₂, which is key for stimulating its proinflammatory response and neurotoxic effect.

It has been suggested that various assembly states of Aβ may coexist in vivo, ranging from monomers and oligomers to protofibrils and fibrils[15]. Despite distinct assembly patterns, the Aβ peptides in monomeric and oligomeric forms mostly have an unstructured N-terminal region, which adopts a flexible conformation[16–18]. In contrast, the Aβ fibrils exhibit a relatively stable structure of the peptide N terminus, which participates in intra- and/or inter-peptide interactions[19,20]. This difference may correlate with the fact that synapse failure and memory impairment can be triggered by the Aβ oligomers but not by the fibrils[21], as the peptide N terminus is required for binding to FPR2, which plays a crucial role in mediating the neurotoxic effects of Aβ. However, this does not rule out the importance of other regions of the peptide in AD pathogenesis, since additional receptors exist for Aβ[15].

**Molecular basis of HN neuroprotection.** It has been reported that HN may exert its neuroprotective effects by competitively inhibiting the access of Aβ42 to FPR2[9], suggesting that Aβ42 and HN share a similar binding pocket in FPR2. Indeed, the FPR2 structures reveal largely overlapped binding sites for these two peptides (Fig. 3a). Similar to Aβ42 and the formyl peptides, fHN binds to the receptor through its N-terminal region, with unambiguous densities for residues fM1–E15 shown in the cryo-EM map (Supplementary Fig. 3d). The C-terminal region of the peptide appears to be flexible and likely lacks contact with the receptor, and thus, was not modeled. This agrees with previous study, which suggested that the C terminus of HN was non-essential for its function because HN and its 21-amino-acid variant (3 residues shorter at the C terminus) had indistinguishable effects[22]. Nevertheless, the N-terminal segment

M1–E15 of HN has not been tested biologically to exclude the requirement of the rest of the peptide for its functionality.

Although having completely different amino-acid sequences, comparison of the fHN- and Aβ42-bound FPR2 structures reveals a similar binding pose for the backbones of the N-terminal segments in fHN (fM1–R4) and Aβ42 (D1–R5) (Fig. 3a). Supported by our functional data, the formylated residue fM1 of fHN plays a crucial role in mediating both receptor activation and receptor–peptide recognition. The agonist potency of fHN was substantially impaired if alanine mutation was introduced in the subpocket that accommodates the peptide N terminus. Especially when the residue D106$^{3.33}$ or V113$^{3.40}$ in helix III was substituted, the fHN-induced cell signaling was abolished (Figs. 1g, 3c; Supplementary Fig. 5b, f; Supplementary Table 3). On the peptide side, the alanine replacement of fM1 resulted in a 200-fold lower binding affinity to the receptor (Fig. 3b and Supplementary Table 1). With a small side chain, the residue A2 in fHN only makes limited contacts with the receptor through its main chain (Fig. 3c). An increased binding affinity observed for the A2W-substituted variant of fHN is consistent with the existence of an empty hydrophobic subpocket adjacent to this residue (Fig. 3b and Supplementary Table 1). In contrast, the neighboring residue P3 packs tightly against helices VI and VII, forming hydrophobic interactions with F257$^{6.51}$ and V284$^{7.35}$ (Fig. 3c). Disrupting this binding interface by mutating either of these two residues to alanine resulted in a 5–72-fold reduction of fHN potency in inducing cell signaling (Figs. 1g, 3d; Supplementary Fig. 5b, j; Supplementary Table 3). Similarly, replacing the peptide residue P3 with alanine decreased the binding affinity by 8-fold (Fig. 3b and Supplementary Table 1).

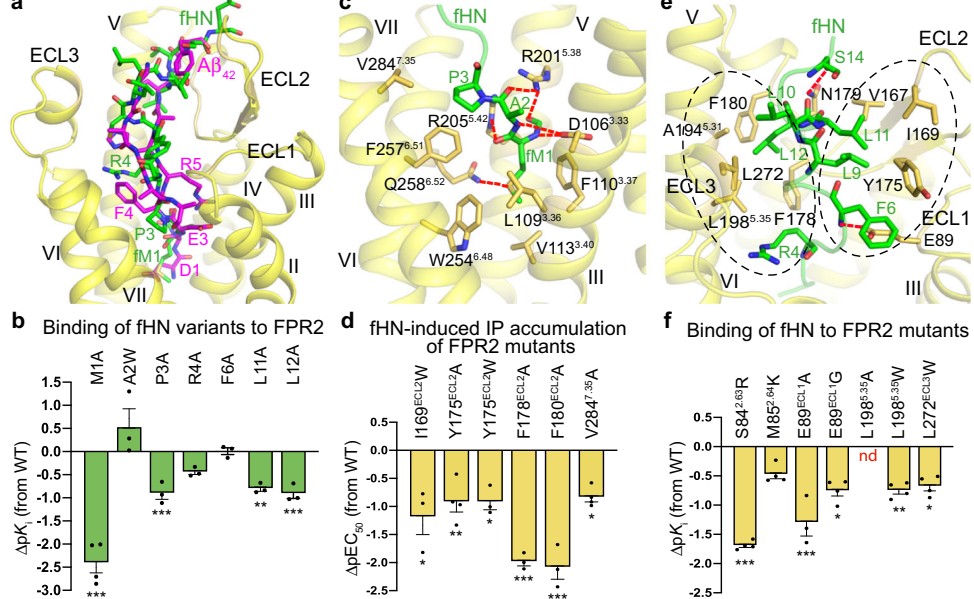

**Fig. 3 Interactions between FPR2 and fHN. a** Comparison of fHN- and Aβ42-binding sites in FPR2. The peptides fHN and Aβ42 are shown as green and magenta sticks, respectively. Only the receptor in the fHN–FPR2–G$_{i2}$ structure is shown in yellow cartoon representation for clarity. **b, d, f** Binding and cell-signaling assays of FPR2. Bars represent differences in calculated fHN-binding affinity (p$K_i$; **b, f**) or potency (pEC$_{50}$, **d**) for each mutant relative to the wild-type receptor or peptide (WT). Data are displayed as mean ± SEM (bars) from at least three independent experiments performed in triplicate with individual data points shown (dots). *$P < 0.05$, **$P < 0.001$, ***$P < 0.0001$ by one-way analysis of variance followed by Dunnett's post-test compared with the response to the wild-type fHN (**b**) or the response of the wild-type receptor (**d, f**). Supplementary Tables 1, 3 provide detailed statistical evaluation, P-values, numbers of independent experiments (n), and expression levels. Source data are provided as a Source Data file. **b** Inhibition of WK (FITC) YMVm binding to wild-type FPR2 by fHN variants. **d** fHN-induced IP accumulation of FPR2 mutants. **f** Inhibition of WK (FITC)YMVm binding to FPR2 mutants by fHN. **c, e** Interactions between FPR2 and fHN. The fHN residues and the receptor residues involved in interactions are shown as sticks. Polar interactions are shown as red dashed lines. **c** Interactions between FPR2 and the fHN residues fM1–P3. **e** Interactions between FPR2 and the fHN residues R4–S14. The two hydrophobic interaction cores are highlighted by two black dashed circles.

Unlike Aβ$_{42}$ that recognizes FPR2 by adopting a β-sheet structure (Fig. 2a), the majority of N-terminal part in fHN (residues R4–E15) exhibits a coiled conformation when bound to the receptor (Fig. 3e). Despite the distinct structures, these two peptides occupy a similar binding site in the extracellular region of the receptor, where the FPR2–fHN recognition is mainly mediated by two hydrophobic interaction cores (Fig. 3a, e). Side chains of R4, L10, and L12 in the peptide interact with a cluster of hydrophobic residues, F178, F180, A194$^{5.31}$, L198$^{5.35}$, and L272, in ECL2, ECL3, and the extracellular tip of helix V, while F6, L9, and L11 bind to a hydrophobic patch in ECL2 that includes V167, I169, and Y175 (Fig. 3e). The critical role of this binding interface in governing HN recognition and function is reflected by an 8–89-fold decrease of the agonistic activity of fHN and/or an over 5-fold reduction of fHN-binding affinity when the receptor residues in this interface were individually replaced with alanine or tryptophan (Fig. 3d, f; Supplementary Fig. 5j, k; Supplementary Tables 1, 3). The more deleterious effect of the substitutions L11A and L12A in fHN (6–8-fold reduction of $K_i$) on FPR2 binding than that of R4A and F6A (<3-fold reduction of $K_i$) is consistent with the fHN–FPR2–G$_{i2}$ structure, in which the leucine-repeat region of the peptide (L9–L12) mediates the most abundant interactions with the receptor in addition to the N-terminal fM1 (Fig. 3e).

To further strengthen the receptor–peptide binding, two polar residues E89 and N179 in the first extracellular loop (ECL1) and ECL2 form hydrogen bonds with main chains of F6 and S14 in fHN, respectively (Fig. 3e). The residue E89 is substituted by glycine in FPR1. Introducing a glycine or alanine at this position in FPR2 decreased the binding affinity of fHN by 6–22-fold (Fig. 3f; Supplementary Fig. 5k; Supplementary Table 1), indicating requirement of this polar residue for the high-affinity binding of fHN and suggesting that this may account for the preference of HN binding to FPR2 over FPR1. The neighboring residues S84$^{2.63}$ and M85$^{2.64}$, of which the counterparts in FPR1 are basic residues R84$^{2.63}$ and K85$^{2.64}$, also contribute to the binding selectivity of HN, as the mutations S84$^{2.63}$R and M85$^{2.64}$K in FPR2 reduced the affinity of fHN by 49-fold and 3-fold, respectively (Fig. 3f; Supplementary Fig. 5k; Supplementary Table 1). The positively charged mutations adjacent to E89 may disturb the polar interaction with the peptide by forming a salt bridge with this acidic residue to constrain its conformation.

The largely overlapped binding sites of Aβ$_{42}$ and fHN offer a molecular basis for HN to competitively block the usage of FPR2 by Aβ$_{42}$ and subsequently impede the Aβ$_{42}$-elicited proinflammatory responses and its fibrillary aggregation. It has been shown that the binding affinity of HN to FPR2 is over 15-fold higher than that of Aβ$_{42}$[23]. With a comparable total binding interface for these two peptides (fHN, 1077 Å$^2$; Aβ$_{42}$, 1166 Å$^2$), the stronger binding of HN, which ensures efficacious blockade of Aβ$_{42}$ binding, is likely gained in two regions. Compared with the residue D1 at the N terminus of Aβ$_{42}$, the longer side chain of M1 in HN makes more extensive contacts with the conserved binding subpocket formed by helices III and VI. In the extracellular region, the residues R5–Y10 of Aβ$_{42}$ exhibit an extended β-strand conformation, interacting with the receptor mainly through the side chains of R5, H6, and Y10 (Fig. 2e). In contrast, the fragment of R4–E15 in fHN adopts a bulkier structure and almost occupies the entire binding groove shaped by the extracellular loops (Fig. 3a, e). This difference results in a larger contribution of the receptor extracellular region (ECL1–3 and residues P187–L198 in the extracellular region of helix V) to fHN binding (48% of binding interface) than Aβ$_{42}$ binding (36% of binding interface).

## Selectivity of formyl peptides.
In addition to the conserved binding site for the N-terminal formylated methionine at the bottom of the ligand-binding pocket, accommodation of the formyl peptides is mainly mediated by two hydrophobic patches in FPR2, which act as two "arms" to stabilize the extended conformation of the peptides (Fig. 4a, b). Residues L/Y2 and I4 in fM5 and fM9 form extensive interactions with ECL2 and helices II, III, and VII of FPR2, while the fM5 residues F3 and I5 and the fM9 residues F3 and I6 face to the opposite direction, making contacts with ECL3 and helices V, VI, and VII (Fig. 4a, b). Having a longer length, fM9 further extends toward the extracellular loops with its C-terminal residue L7 forming additional hydrophobic interactions with ECL2 (Fig. 4c). The polar residue N5 forms a hydrogen bond with E89 in ECL1 (Fig. 4c). Likely due to lack of interactions with the receptor, no clear densities were observed for the last two residues of fM9. The binding pattern of the fMLF residues L2 and F3 at FPR1 aligns well with that of the residues at positions 2 and 3 in the formyl peptides at FPR2 (Fig. 4d). Owing to a spatial hindrance caused by the larger side chains of residues F81$^{2.60}$ and Y257$^{6.51}$ in FPR1 (L81$^{2.60}$ and F257$^{6.51}$ in FPR2) and a different orientation of Y257$^{6.51}$, fMLF moves toward helices IV and V, excluding its contact with helix VII (Fig. 4d).

Despite high sequence identity (69%), FPR1 and FPR2 display distinct binding behaviors to the formyl peptides. The *E. coli*-derived chemotactic peptide fMLF exhibits full agonistic activity at FPR1, but is a weak agonist for FPR2 with an over 400-fold lower binding affinity[24]. The FPR1 residues involved in fMLF binding are highly conserved in FPR2, except for F81$^{2.60}$, F102$^{3.29}$, and Y257$^{6.51}$ (L81$^{2.60}$, H102$^{3.29}$, and F257$^{6.51}$ in FPR2). The role of these residues in determining ligand selectivity was investigated by mutagenesis studies. However, the single FPR1-to-FPR2 swap mutations, F81$^{2.60}$L, F102$^{3.29}$H, and Y257$^{6.51}$F, and their combination showed little effect on the binding affinity of fMLF to FPR1 (Supplementary Fig. 5l and Supplementary Table 1). These data suggest that the interactions between the receptor and the peptide in the final binding pose likely do not contribute much to the ligand selectivity.

Further inspection of the ligand-binding pockets in FPR1 and FPR2 reveals distinct charge distribution. The narrow entrance to the ligand-binding pocket in FPR1, which is adjacent to the C terminus of fMLF, is positively charged due to residues R84$^{2.63}$ and K85$^{2.64}$ at the extracellular tip of helix II (Fig. 4e). Although these basic residues do not form any direct contact with the tripeptide, they may provide an anchor for the C-terminal negatively charged carboxyl group of the peptide during initial receptor–peptide recognition to facilitate the entry of the peptide into the deep ligand-binding cavity. In contrast, in FPR2, these two residues are replaced with uncharged residues S84$^{2.63}$ and M85$^{2.64}$. Instead, the entrance to the binding pocket is negatively charged with E89 and D281$^{7.32}$, counterparts of which are glycines in FPR1 (Fig. 4f). These two acidic residues may act as a "shield" to repel the negative charge at the C terminus of fMLF and limit the peptide entry. The importance of the different charges in defining the peptide selectivity was supported by previous mutagenesis studies, showing that the FPR1 mutants R84$^{2.63}$A and K85$^{2.64}$A had a reduced binding affinity to [$^3$H]fMLF[25], while the mutations S84$^{2.63}$R, M85$^{2.64}$K, and E89G of FPR2 showed an increased binding of [$^3$H]fMLF[26]. The negative charge in the FPR2 ligand-binding pocket is also consistent with the fact that a basic residue at the C terminus of a formylated tetrapeptide (fMLFK) is more favorable for binding to FPR2 than an acidic residue (fMLFE)[24].

## Promiscuous ligand recognition of FPR2.
FPR2 is well known for its promiscuity and versatility in sensing a variety of pathogen- and host-derived peptides with very limited sequence similarities[1]. Opposed ligand-binding preferences of FPR1 and FPR2 have been demonstrated by previous investigation using

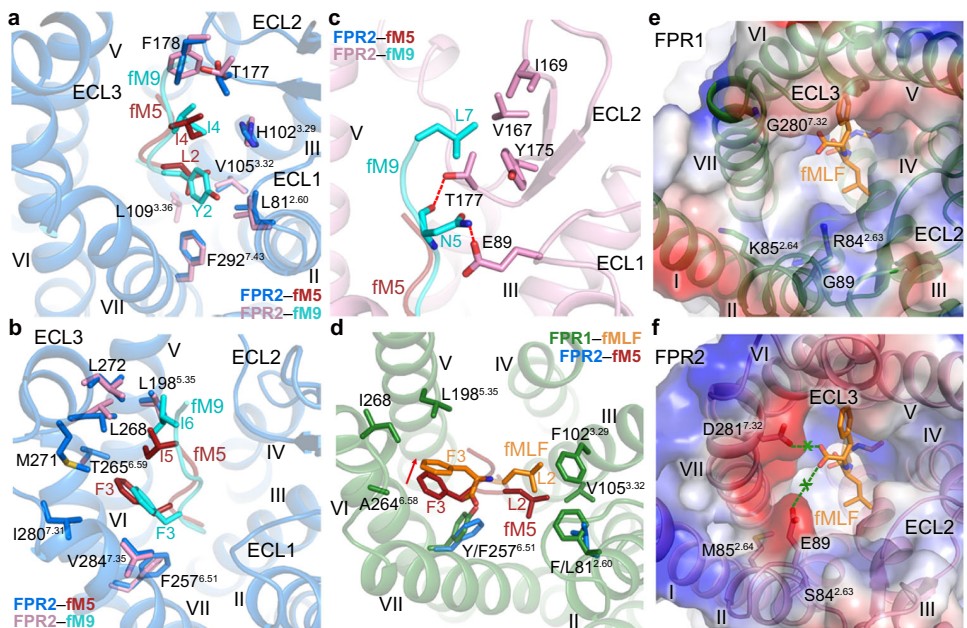

**Fig. 4 Binding modes of formyl peptides at FPR1 and FPR2. a–c** Interactions between FPR2 and the formyl peptides. Only the receptor in the fM5–FPR2–G$_{i2}$ (**a**, **b**) or fM9–FPR2–G$_{i2}$ structure (**c**) is shown in cartoon representation for clarity. **a** Interactions between FPR2 and the residues Y/L2 and I4 in fM5 and fM9. **b** Interactions between FPR2 and the residues F3 and I5/I6 in fM5 and fM9. **c** Interactions between FPR2 and the residues N5 and L7 in fM9. Polar interactions are shown as red dashed lines. **d** Interactions between FPR1 and the residues L2 and F3 in fMLF. The peptide fM5 and FPR2 residues L81$^{2.60}$ and F257$^{6.51}$ in the fM5–FPR2–G$_{i2}$ structure are also shown for comparison. The red arrow indicates the movement of fMLF relative to fM5. **e**, **f** Surfaces of the FPRs are colored according to their electrostatic potential from red (negative) to blue (positive), showing different charge distributions at the entrance to the ligand-binding pocket in the two receptors. **e** The peptide fMLF and the receptor residues R84$^{2.63}$, K85$^{2.64}$, G89, and G280$^{7.32}$ in the fMLF–FPR1–G$_{i1}$ structure are shown as sticks. **f** The receptor residues S84$^{2.63}$, M85$^{2.64}$, E89, and D281$^{7.32}$ in the fM9–FPR2–G$_{i2}$ structure are shown as sticks. The peptide fMLF in the fMLF–FPR1–G$_{i1}$ structure is also shown. The green "x" indicates that the negatively charged C terminus of fMLF repels the acidic residues E89 and D281$^{7.32}$ in FPR2.

staphylococci-produced formyl peptides and variants, showing that short peptides were strong FPR1 agonists but the potency declined with increasing peptide length, whereas long peptides (longer than 18 residues) favor FPR2 activation and the activity increased with length[27]. Consistent with this finding, both Aβ$_{42}$ and HN, which contain 42 and 24 residues, respectively, display much higher agonistic activity at FPR2 than at FPR1[23]. These data suggest that peptide length is crucial for FPR stimulation and raise the question about the molecular factor that governs the different peptide-recognition patterns of FPR1 and FPR2.

The structures of FPR2 in complex with various peptides reveal a charged binding cavity within the transmembrane helical bundle that accommodates the polar N termini of the peptides and a binding groove on the receptor extracellular side that interacts with the peptides mainly through hydrophobic interactions. The Aβ$_{42}$- and fHN-bound FPR2 structures, together with functional data, demonstrate the importance of the receptor extracellular region in recognizing the long peptides. A similar conformation of the receptor extracellular region is shared by all the FPR2 structures solved in the present study, despite the diverse sequences and lengths of the peptide ligands bound (Supplementary Fig. 4d). This suggests that the largely open extracellular region of FPR2 is subtype-specific, providing space to accommodate the long peptides and/or facilitating access of the large ligands into the narrow binding cavity deep in the helical bundle. By comparing the peptide–FPR2–G$_{i2}$ structures with the fMLF–FPR1–G$_{i1}$ structure, conformational differences were observed in the extracellular parts of the receptors (Supplementary Fig. 4e). The extracellular tip of helix V and ECL3 in FPR1 move inward by 4 Å (measured at Cα of P187) and 6 Å (measured

at Cα of G/Y274), respectively, resulting in a less-open ligand-binding pocket in FPR1 (Fig. 1b; Supplementary Fig. 4e–g). The inward positioning of helix V and ECL3 would form a steric clash with the large peptides and limit their entry (Supplementary Fig. 4h).

The large opening of the ligand-binding pocket in FPR2 provides spatial redundancy, allowing recognition of peptides with diverse sequences and structures. The requirement of abundant space in the extracellular region for FPR2 binding to long peptides is also supported by our mutagenesis studies. Reducing the size of the ligand-binding pocket by introducing a tryptophan mutation in ECL2 (I169W or Y175W), ECL3 (L272W), or the extracellular region of helix V (L198$^{5.35}$W) substantially impaired fHN binding or cell signaling induced by Aβ$_{42}$ and fHN (Fig. 3d, f; Supplementary Fig. 5i–k; Supplementary Tables 1, 3). These structural and functional data suggest that the conformational difference in the extracellular regions of FPR1 and FPR2 may be a determinant of their ligand preferences, in addition to the distinct charge distributions within the receptor helical bundles that govern recognition of the polar N termini of the peptides. This finding provides premise of the FPR2 extracellular region as a potential drug-binding site, offering new opportunities for drug development for the treatment of AD and other inflammatory diseases.

## Methods
**Construct design.** The genes of human FPR1 and FPR2 were codon-optimized and synthesized by Sangon Biotech for insect-cell expression. The FPR1 gene was then cloned into a modified pFastBac1 vector containing an expression cassette with an N-terminal hemagglutinin (HA) signal sequence and a PreScission

protease site followed by a 2 × Strep tag and a Flag tag at the C terminus. The C-terminal residues R322–K350 were truncated to improve protein yield and homogeneity. A dominant negative Gα$_{i1}$ subunit (DNGα$_{i1}$) was generated by introducing five mutations, S47C, G202T, G203A, A326S, and E245A, to increase stability of the G$_{i1}$ heterotrimer. The human Gβ$_1$ with a 6 × His tag at the N terminus and Gγ$_2$ were cloned into the pFastBac Dual vector (Invitrogen). The codon-optimized DNA sequences and all primer sequences used in this study are included in Supplementary Table 4. FPR1, DNGα$_{i1}$, Gβ$_1$, and Gγ$_2$ were coexpressed in HighFive insect cells (Invitrogen) using the Bac-to-Bac Baculovirus Expression System (Invitrogen). Cells were grown to a density of $1.5 \times 10^6$ cells per ml at 27 °C and infected with high-titer viral stocks at an MOI (multiplicity of infection) ratio of 1:1:1 for FPR1, DNGα$_{i1}$, and Gβ$_1$γ$_2$. Cells were then harvested by centrifugation at 48 h post infection and stored at −80 °C for further use.

The human FPR2 gene was cloned into a modified pFastBac1 vector with the HA signal peptide at the N terminus and the PreScission protease site followed by the Flag tag and 2 × Strep tag at the C terminus. The fusion protein b$_{562}$RIL (PDB ID: 1M6T) followed by a TEV (tobacco etch virus) protease-cleavage site was connected to the N terminus of FPR2. The C-terminal residues E347–M351 were truncated and a mutation S211$^{5.48}$L was introduced to improve protein quality. A dominant negative Gα$_{i2}$ (DNGα$_{i2}$) was generated by introducing four mutations, S47N, G204A, A327S, and E246A. The modified FPR2, DNGα$_{i2}$, and Gβ$_1$γ$_2$ were coexpressed in HighFive insect cells using the Bac-to-Bac Baculovirus Expression System. Cells were grown to a density of $1.5 \times 10^6$ cells per ml and then infected with high-titer viral stocks ($>10^9$ viral particles per ml) at an MOI ratio of 7:4:4 for FPR2, DNGα$_{i2}$, and Gβ$_1$γ$_2$. The cells were cultured at 27 °C for 48 h and then harvested by centrifugation and stored at −80 °C.

**Purification of fMLF–FPR1–G$_{i1}$ complex.** The cell pellets from 500 ml of cell culture were thawed and suspended in 50 ml of suspension buffer containing 25 mM HEPES, pH 7.5, 150 mM NaCl, 10 mM MgCl$_2$, 10% (v/v) glycerol, and EDTA-free protease-inhibitor cocktail tablets (Roche). The suspension was further supplemented with 50 μM fMLF (GL Biochem) and 25 mU mL$^{-1}$ apyrase (New England BioLabs), and incubated at 20 °C for 1 h. Cell membranes were then collected by centrifugation at 160,000 g for 30 min. The membranes were resuspended and solubilized in 50 ml of solubilization buffer containing 25 mM HEPES, pH 7.5, 150 mM NaCl, 10 mM MgCl$_2$, 10% (v/v) glycerol, 25 mU ml$^{-1}$ apyrase, 0.5% (w/v) n-dodecyl-β-D-maltopyranoside (DDM) (Anatrace), 0.1% (w/v) cholesterol hemisuccinate (CHS) (Sigma), and 50 μM fMLF at 4 °C for 2 h. The solubilized fraction was isolated by centrifugation at 160,000 g for 30 min and then incubated with 1 ml of Strep-Tactin Sepharose (IBA Lifesciences) at 4 °C overnight.

The resin with immobilized complex protein was washed with fifteen column volumes of washing buffer 1 containing 25 mM HEPES, pH 7.5, 150 mM NaCl, 10 mM MgCl$_2$, 0.05% (w/v) DDM, 0.01% (w/v) CHS, and 25 μM fMLF. Then the detergent was exchanged by incubating the resin in 25 mM HEPES, pH 7.5, 150 mM NaCl, 10 mM MgCl$_2$, 0.25% (w/v) glyco-diosgenin (GDN) (Anatrace), and 50 μM fMLF at 4 °C for 2 h. After that, the resin was washed with ten column volumes of washing buffer 2 containing 25 mM HEPES, pH 7.5, 150 mM NaCl, 10 mM MgCl$_2$, 0.01% (w/v) GDN, and 25 μM fMLF. The complex protein was then eluted with 5 column volumes of 200 mM Tris-HCl, pH 8.0, 150 mM NaCl, 10 mM MgCl$_2$, 50 mM biotin, 0.01% (w/v) GDN, and 50 μM fMLF. For further purification, the protein sample was concentrated to 500 μl using a 100-kDa molecular-weight cutoff concentrator (Millipore) and subjected to size-exclusion chromatography using a Superdex 200 Increase 10/300 column (GE Healthcare) preequilibrated with a buffer containing 25 mM HEPES, pH 7.5, 150 mM NaCl, 10 mM MgCl$_2$, 0.01% (w/v) GDN, and 5 μM fMLF. Finally, the purified complex was concentrated to 3.5–6 mg ml$^{-1}$ using a 100-kDa molecular-weight cutoff concentrator (Millipore) for cryo-EM experiments. The purity and homogeneity of the complex were analyzed by SDS-PAGE, native PAGE, and analytical size-exclusion chromatography using a 4.6 × 250 mm Nanofilm SEC-250 column (Sepax Technologies).

**Purification of peptide agonist–FPR2–G$_{i2}$ complexes.** The cell pellets from 200 ml of cell culture that expresses the FPR2–G$_{i2}$ complex were thawed on ice and suspended in 20 ml of lysis buffer containing 20 mM HEPES, pH 7.5, 50 mM NaCl, 2 mM MgCl$_2$, 25 mU ml$^{-1}$ apyrase, TEV protease (custom-made), and EDTA-free protease-inhibitor cocktail tablets, and incubated at room temperature for 1 h. Then the lysate was homogenized using a dounce homogenizer. The cell membranes were collected by centrifugation at 160,000 g for 30 min, and resuspended in 20 ml of resuspension buffer containing 50 mM HEPES, pH 7.5, 150 mM NaCl, 10 mM MgCl$_2$, 25 mU ml$^{-1}$ apyrase, and EDTA-free protease-inhibitor cocktail. The peptide–FPR2–G$_{i2}$ complex was formed in the presence of 50 μM fM5, fM9, fHN, or Aβ$_{42}$ (GL Biochem), and extracted from the membranes by incubating with 0.5% (w/v) lauryl maltoseneopentyl glycol (LMNG) (Anatrace) and 0.1% (w/v) CHS at 4 °C for 3 h. The solubilized fraction was collected by ultracentrifugation at 40,000 g for 30 min and incubated with 400 μl of Strep-Tactin Sepharose at 4 °C overnight.

The resin was washed with 20 column volumes of washing buffer containing 20 mM HEPES, pH 7.5, 150 mM NaCl, 2 mM MgCl$_2$, 0.01% (w/v) LMNG, 0.001% (w/v) CHS, and 25 μM peptide ligand (fM5, fM9, fHN, or Aβ$_{42}$). The complexes were eluted with 5 column volumes of elute buffer containing 150 mM Tris-HCl, pH 8.0, 150 mM NaCl, 2 mM MgCl$_2$, 50 mM biotin, 0.01% (w/v) LMNG, 0.001% (w/v) CHS, and 50 μM peptide ligand. The protein samples were then concentrated to 500 μl using a 100-kDa molecular-weight cutoff concentrator (Millipore) and subjected to size-exclusion chromatography using a Superdex 200 Increase 10/300 column (GE Healthcare) preequilibrated with 20 mM HEPES, pH 7.5, 150 mM NaCl, 0.01% (w/v) LMNG, 0.001% (w/v) CHS, and 5 μM peptide ligand. The purified complexes were concentrated to 1.5 mg ml$^{-1}$ with a 100-kDa molecular-weight cut-off concentrator (Millipore), and analyzed by SDS-PAGE and analytical size-exclusion chromatography using a 4.6 × 250 mm Nanofilm SEC-250 column (Sepax Technologies).

**Cryo-EM data acquisition and processing.** The fMLF–FPR1–G$_{i1}$ complex was diluted to 1.5 mg ml$^{-1}$ using a buffer containing 25 mM HEPES, pH 7.5, 150 mM NaCl, 10 mM MgCl$_2$, and 0.01% (w/v) GDN. The G$_{i2}$-bound FPR2 complexes were diluted to 1 mg ml$^{-1}$ using a buffer containing 20 mM HEPES, pH 7.4, 150 mM NaCl, 0.01% (w/v) LMNG, and 0.001% (w/v) CHS. Then 3 μl of protein sample was applied to glow-discharged holey grids (CryoMatrix R1.2/1.3, Au 300 mesh) and vitrified at 4 °C and 100% humidity with blot time of 0.5 s and blot force of 0 using a Mark IV Vitrobot (ThermoFisher Scientific), followed by flash-frozen in liquid ethane. Cryo-EM images were collected on a 300 kV Titan Krios G3 electron microscope (FEI) equipped with K3 summit direct-detection camera (Gatan) and a GIF-Quantum LS Imaging energy filter with a slit width of 20 eV. The super-resolution counting mode of SerialEM program[28] was used to capture movies automatically with a pixel size of 1.045 Å. Movie stacks were recorded with the defocus values varying from −0.8 μm to −1.5 μm and generated by 3-s exposure with 32 frames. The dose rate was 2.1875 electrons per Å$^2$ per frame.

For the fMLF–FPR1–G$_{i1}$ complex, two datasets were collected and individually subjected to motion correction, autopicking, two-dimensional (2D) classification, and three-dimensional (3D) classification. For the first dataset, 4284 movies were collected and subjected to a beam-induced motion correction using MotionCor2[29]. Gctf software[30] was used to determine contrast-transfer function (CTF) parameters for each image. Guided by a template generated from manual picking, autopicking in RELION-3[31] was performed to extract particle projections. In all, 3,090,168 particles were extracted for 2D and 3D classification. In all, 982,613 particles of the best-looking class were subjected to 3D refinement, resulting in a map at 4.0 Å resolution. For the second dataset, 3310 movies were collected and similarly processed. In all, 2,973,334 particles were extracted for 2D and 3D classification. Then 833,920 particles of the best-looking class were selected for 3D refinement, generating a 4.0-Å resolution map. The two datasets were subsequently merged and subjected to Bayesian polishing, another round of 3D classification and 3D refinement with RELION-3, isolating a final partition of 1,299,041 particles and resulting in a final 3.3-Å map with a B factor of −135 Å$^2$.

A total of 5347 movies of the fM5–FPR2–G$_{i2}$ complex were collected and subjected to a beam-induced motion correction using MotionCor2[29]. Gctf software[30] was used to determine CTF parameters for each image. Guided by a template generated from manual picking, autopicking in RELION-3[31] was performed to extract particle projections. In total, 5,637,810 particles were extracted for 2D and 3D classification. In all, 1,216,356 particles of the best-looking classes were subjected to Bayesian polishing and 3D autorefinement using RELION-3, resulting in a 2.9-Å map with a B factor of −97 Å$^2$.

For the fM9–FPR2–G$_{i2}$ complex, a total of 4,302 movies were collected and subjected to a beam-induced motion correction using MotionCor2[29]. Gctf software[30] was used to determine CTF parameters for each image. Guided by a template generated from manual picking, autopicking in RELION-3[31] was performed to extract particle projections. In total, 2,379,340 particles were extracted for 2D and 3D classification. Then 503,133 particles of the best-looking class were selected for 3D refinement to 3.2 Å resolution. For further refinement, 422,636 particles were selected by another round of 3D classification after Bayesian polishing, and then subjected to 3D refinement using RELION-3, resulting in a 3.1-Å map with a B factor of −85 Å$^2$.

Two datasets of the fHN–FPR2–G$_{i2}$ complex were collected and individually processed. In the first dataset, 5,184 movies were collected and subjected to a beam-induced motion correction using MotionCor2[29]. Gctf software[30] was used to determine CTF parameters for each image. Guided by a template generated from manual picking, autopicking in RELION-3[31] was performed to extract particle projections. In all, 4,135,001 particles were extracted for 2D and 3D classification. In all, 900,679 particles of the best-looking class were subjected to 3D refinement, resulting in a map at 3.2 Å resolution. In the second dataset, 3512 movies were collected and processed as above. In all, 5,376,740 particles were extracted for 2D and 3D classification. Then, 985,984 particles of the best-looking class were selected for 3D refinement to 3.3 Å resolution. The two datasets were subsequently merged and subjected to Bayesian polishing, another round of 3D classification and 3D refinement with RELION-3, isolating a final partition of 1,398,841 particles and resulting in a final 2.8-Å map with a B factor of −101 Å$^2$.

For the Aβ$_{42}$–FPR2–G$_{i2}$ complex, two datasets were collected and individually processed. For the first dataset, 3534 movies were collected and subjected to a beam-induced motion correction using MotionCor2[29]. Gctf software[30] was used to determine CTF parameters for each image. Guided by a template generated from

manual picking, autopicking in RELION-3[31] was performed to extract particle projections. In all, 2,598,541 particles were extracted for 2D and 3D classification. In all, 393,979 particles of the best-looking classes were subjected to 3D refinement, which produced a map with an overall resolution of 3.5 Å. The second dataset with 4332 movies was processed as above. In all, 3,421,506 particles were extracted for 2D and 3D classification. Then 1,175,437 particles of the best-looking class were selected to 3D refinement to 3.0 Å resolution. The two datasets were subsequently merged and subjected to Bayesian polishing, another round of 3D classification and 3D refinement with RELION-3, isolating a final partition of 1,094,657 particles and resulting in a final map at 3.0 Å resolution with a $B$ factor of −96 Å$^2$.

All the reported resolutions were determined using gold-standard Fourier shell correlation (FSC) with the 0.143 criteria. Local resolution was determined using ResMap[32].

**Model building and refinement**. The models of the peptide agonist–FPR–$G_i$ complexes were built using the $G_i$ heterotrimer from the μ-opioid receptor (μOR)–$G_i$ complex structure (PDB ID: 6DDE) and the FPR2–WKYMVm crystal structure (PDB ID: 6LW5) as initial models. All the models were docked into the cryo-EM electron-density maps using Chimera[33], followed by iterative manual adjustments in COOT[34] and real-space refinement using phenix.real_space_refine in Phenix[35].

The final model of fMLF–FPR1–$G_{i1}$ contains 298 residues of FPR1 (S19–L316) and 3 residues of fMLF (fM1–F3). The final models of fM5–FPR2–$G_{i2}$ and Aβ$_{42}$–FPR2–$G_{i2}$ contain 298 residues of FPR2 (S19–S316), 5 residues of fM5 (fM1–I5), and 16 residues of Aβ$_{42}$ (D1–Y10 and G37–A42). The final models of fM9–FPR2–$G_{i2}$ and fHN–FPR2–$G_{i2}$ contain 299 residues of FPR2 (S19–L317), 7 residues of fM9 (fM1–L7), and 15 residues of fHN (fM1–E15). The remaining residues of FPR1, FPR2, and ligands are disordered and were not modeled. The models were validated using Molprobity[36]. Structural figures were prepared by Chimera or PyMOL (https://pymol.org/2/). The data-collection and structure-refinement statistics are provided in Supplementary Table 2.

**Ligand-binding assay**. The genes of wild-type FPR1 and FPR2 and mutants with a Flag tag at the N termini were cloned into pTT5 vector (Invitrogen) and expressed in HEK293F cells (Invitrogen). Cells were harvested 48 h post transfection with 1 μg ml$^{-1}$ plasmid. Cell-surface expression of the receptors was measured by mixing 10 μl cells with 15 μl of TBS buffer supplemented with 4% BSA, 20% (v/v) viability-staining solution 7-AAD (Invitrogen, Cat#00-6993-50), and 0.1% (v/v) anti-FLAG M2-FITC antibody (Sigma, Cat#F4049) at 4 °C for 20 min. After incubation, 175 μl of TBS buffer was added and the fluorescent signal was measured using a flow-cytometry reader (Guava easyCyte HT, Millipore). Then, the cells were washed and resuspended to a final concentration of $1 \times 10^6$ cells per ml in Hanks' Balanced Salt Solution (HBSS) buffer supplemented with 0.5% bovine serum albumin (BSA) and 20 mM HEPES, pH 7.4.

For saturation binding, the cells were plated in 96-well plates (100,000 cells per well) and incubated with different concentrations of fluorescein isothiocyanate (FITC)-conjugated peptide WK(FITC)YMVm (2 nM–250 nM) at 4 °C for 1 h. The mean fluorescent intensity of each well was then read by the FCM reader. Total binding and nonspecific binding were measured in the absence and presence of the unlabeled ligand (200 μM WKYMVm), respectively. For competition binding, the cells were incubated with 10 nM WK(FITC)YMVm at 4 °C for 1 h. Increasing concentrations of different ligands were then added (fMLF (for FPR1) and fM5, fM9, and Aβ (for FPR2), 100 pM–1 mM; fHN and variants (for FPR2), 10 pM–100 μM) and incubated at 4 °C for another 1 h. Mean fluorescent-intensity values were measured by flow cytometry. Data were analyzed using Prism 8.0 (GraphPad software).

**Inositol-phosphate (IP) accumulation assay**. Flag-tagged wild-type FPR1 and FPR2 and mutants were cloned into the pTT5 vector and expressed in HEK293F cells. Cells were cotransfected with the plasmids of receptor and a chimeric Gα protein Gα$_{\Delta6qi4myr}$, which redirects the Gα$_i$ signaling pathway to the Gα$_q$ phospholipase-C pathway[37], at a ratio of 1:2 (w/w), and were harvested 48 h post transfection. The cell-surface expression was measured as described above.

The IP-One Gq assay kit (Cisbio Bioassays, 62IPAPEB) was used to measure the peptide agonist-induced IP production. The cells were plated in 384-well plates (20,000 cells per well) and incubated with different concentrations of peptide agonist (fMLF, fM5, fM9, and fHN, 10 pM–100 μM; Aβ$_{42}$ and variants, 100 pM–1 mM) at 37 °C for 90 min. Then, the cells were supplemented with 3 μl cryptate-labeled anti-IP1 monoclonal antibody (1:20 diluted in lysis and detection buffer) and 3 μl d2-labeled IP1, and incubated at room temperature for another 1 h. Fluorescent signal was measured using an EnVision multilabel-plate reader (PerkinElmer) with excitation at 330 nm and emission at 620 nm and 665 nm. The accumulation of IP1, EC$_{50}$, and pEC$_{50}$ were calculated using nonlinear regression (curve fit) according to a standard dose–response curve in GraphPad Prism 8.0 (GraphPad software).

**Reporting summary**. Further information on research design is available in the Nature Research Reporting Summary linked to this article.

## Data availability

Atomic coordinates and cryo-EM-density maps for the structures of fMLF–FPR1–$G_{i1}$, fM5–FPR2–$G_{i2}$, fM9–FPR2–$G_{i2}$, fHN–FPR2–$G_{i2}$, and Aβ$_{42}$–FPR2–$G_{i2}$ complexes have been deposited in the RCSB Protein Data Bank (PDB) under accession codes 7WVU, 7WVV, 7WVW, 7WVX, and 7WVY, and the Electron Microscopy Data Bank (EMDB) under accession codes EMD-32858, EMD-32859, EMD-32860, EMD-32861, and EMD-32862. The database used in this study includes PDB 1M6T, 6LW5, 6C1R, and 6DDE. All relevant data are available from the corresponding authors upon reasonable request. Source data are provided with this paper.

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

## Acknowledgements

The cryo-EM studies were performed at the EM facility of Shanghai Institute of Materia Medica (SIMM), Chinese Academy of Sciences. We thank Q. Wang from SIMM for cryo-EM data collection. This work was supported by the National Science Foundation of China grants 31730027, 31825010, and 82121005 (B.W.), National Key R&D Program of China 2018YFA0507000 (B.W. and Q.Z.), CAS Strategic Priority Research Program XDB37030100 (B.W. and Q.Z.), and Shanghai Pilot Program for Basic Research—Chinese Academy of Sciences, Shanghai Branch JCYJ-SHFY-2021-008 (B.W.).

## Author contributions

Y.Z. solved the FPR structures, designed the functional assays, and wrote the first draft. X.L. and X.Z. developed the protein expression and purification procedures, prepared the protein samples for cryo-EM, and performed the functional assays. S.H. and M.W. prepared cryo-samples, collected cryo-EM data, and performed cryo-EM data processing and analysis. Y.S. helped with protein preparation and functional assays. L.M., X.C., and C.Y. expressed the proteins. Q.Z. and B.W. initiated the project, planned and analyzed experiments, supervised the research, and wrote the paper with input from all co-authors.

## Competing interests

The authors declare no competing interests.
