## [Peer Review File · Nature Communications]

REVIEWER COMMENTS

Reviewer #1 (Remarks to the Author):

This is a comprehensive cryo-EM study of FPR2 with peptide ligands including A β 42, the neuroprotective peptide humanin, and two smaller peptides in addition to a FPR1 and small peptide. Residues responsible for affinity and signaling for FPR2 and A β 42 and humanin and, and selectivity with respect to FPR1, a GPCR with ~70% identity to FPR2 are verified by mutations and IP assays. This study follows the X-ray study by Wu et al in Nat Commun on FRP-2 last year with the peptide agonist WKYMVm last year. The described interactions of FPR2 with peptides of various sequences and lengths provides insight into its binding promiscuity relative FPR1 and, significantly, could provide an understanding the AD pathology with the structure of A β 42 and the foundation for structure-based drug design. All five structures were determined with the Gai β 1 γ 2. Since there is an allosteric relationship between the orthosteric and G-protein binding sites, the authors should discuss if there are any differences in the interactions between FPRs-Gai for all five ligands and, if there are, to describe them. They comment that helices VI and VII move to accommodate Gai, and the structures of FPR and G-proteins align well, but there are no comment/discussion of side chain interactions for each agonist. For A β 42, it is internalized and interactions with arrestin might be more relevant for AD, but the absence of any discussion with Gai for all ligands in FPRs is a missed opportunity. At the very least, they should state they do not observe any differences in the interactions (and distances) among the side chains of Gai and receptors, if this is the case.

Minor corrections:

Line 60, "stays unknown" should be "is unknown"

Line 93, "Structural alignment of the active FPR structures..." should be "Alignment of the active FPR structures..."

Line 200, although implied by the last sentence on this line, the authors should make a statement to note the residues of humanin peptide defined by density (1-15) have not been tested biologically.

Line 220, "the" should precede "majority"

Line 249, the authors provide a structural explanation for the 15-fold greater affinity for humanin versus A β 42 and provide the percentage of the binding interface for only the extracellular loops. They should also provide the total binding interface for humanin and A β 42 as they note that humanin makes "extensive contacts" compared to A β 42 in the subpocket.

Line 271, I don't know of a case where "extracellular milieu" in this context. They should change it to "extracellular loops."

Line 276, Y257(6.51) in FPR1 not only has a hydroxyl group (compared to F257(6.51) in FPR2) but there is slightly different orientation that would clash with fMLF. The authors should make a note of this.

Line 344, "promise" to be "premise"

Line 428 and 438, please explain what is meant by "technical" triplicate. Do you mean "at least in triplicate?" In addition for line 438, why was A β 42 performed with n=18 and the

others with an n=3?

Figure 3, The superposition in (a) is not useful. It does not show the β -strand vs the coiled structure for the peptides. I recommend another two separate figures shown the cartoon form. These two cartoon forms could be here or somewhere in Fig. 1.

Line 536, there should "the" between "in" and "presence"

Reviewer #2 (Remarks to the Author):

The manuscript from Zhu et al shows one cryo-EM structure of the formyl peptide receptor 1 (FPR1) and 4 structures of FPR2 with different peptide agonists bound including the beta amyloid peptide Abeta42 that forms plaque in the brain of Alzheimer's patients. The manuscript has provided detailed insights in the ligand binding and recognition in FPRs. The authors also discovered the difference between FPR1 and FPR2 that resides at the extracellular region, where FPR1 shows a narrower entry for ligands than FPR2. Besides the cryo-EM structures, the authors provide thorough and comprehensive mutational studies to identify the impact of the residues that are involved in binding and also the signalling activity. Overall, this is an outstanding manuscript.

As the authors are linking FPR2 with Alzheimer's disease by solving the structure of FPR2-Gi-Abeta42, I only have a few points around Abeta40/42 that may require a bit more study and/or citation.

1) Line 137-139: "The other parts of Abeta42 may play a role in stabilizing the conformation of the peptide N terminus to enable FPR2 binding and/or is key for initial receptor-peptide recognition." I suggest the authors to mention the folding of Abeta42, as Abeta42 can fold into a variety of structures, and apparently only the specific form of Abeta42 can activate/bind FPR2.

2) The authors show the binding of Abeta42 and Abeta40 at more or less the same level (Fig. 2b). In the solved structure, Abeta42 residues 1-10 and 37-42 form an antiparallel beta-sheet. What about Abeta40? Would Abeta40 also form such an antiparallel beta-sheet and activate FPR2 as Abeta42? Does Abeta40 also work as agonist or partial agonist or antagonist/inverse agonist? This question is raised because Abeta42 is more toxic than Abeta40 in Alzheimer's disease. Therefore, it would be great to address if FPR2 can differentiate Abeta40 and Abeta42 in signalling. As the reviewer, it is unreasonable to ask the authors to solve one more structure of FPR2-Gi-Abeta40 (if it works), but it would be good to show the signalling activity of FPR2-Abeta40 in comparison with FPR2-Abeta42 by IP accumulation study as in Extended Figure 5.

Below are minor points:

3) Line 135-137: "These data suggest that the very C terminus of Abeta42 does not contribute much to its binding affinity to FPR2, ...". Maybe change the "very C terminus" to "the last two amino acids", as it sort of leads the reader to look for Abeta37-42 binding assay in Fig. 2b.

4) Line 184: "corelate". Please change to "correlate".

5) Line 492-3: Is the HA signal peptide at the C-terminus? From the written text, it looks like HA is after the C-terminus of FPR2.

6) Line 504-5/530-1: Please describe the ratio between the cell pellet (gram) and

suspension buffer (mL).

7) Line 508/534-5: Please describe the ratio between membrane (gram or mg/mL) and resuspension buffer.

8) Line 512/539: "The supernatant was...": Please change "supernatant" to "solubilized fraction".

9) Line 513/540: Please describe how much resin is used for the material from ___ gram of cell pellet.

10) Line 521: Is it 200 mM Tris-HCl or 20 mM Tris-HCl?

11) Line 522/550: Between Strep-tag purification and SEC, is there any concentration step introduced?

12) Line 526: Please describe the concentrator used.

13) Line 527: Please describe the column used for aSEC, here or in Extended Figure 1 legend.

14) Line 533: using a "Dounce homogenizer".

15) Line 562: Please add "Angstrom" to the pixel size.

16) Extended Data Fig. 1a/b/c: Please describe ligands used here.

Suggestions for the future work:

A) The authors show that Abeta42 residues 1-10 and 37-42 form an antiparallel beta-sheet (Fig. 2a), and also Abeta12 alone is not enough to bind FPR2 (Fig. 2b). This is a very important and remarkable finding. Have the authors tested mixing the N-terminal peptide Abeta1-12 and the C-terminal Abeta37-42 and see if such a mixture can activate FPR2? If N-terminal or C-terminal peptides alone are not enough to activate FPR2 but requires the synergy from both terminal peptides, then it will demonstrate the distinct function of Abeta42 from other Abeta peptides in causing AD.

B) Line 321-322: "..., demonstrating the importance of the receptor extracellular region in recognizing the long peptides." It would be great to see if swapping the ECLs between FPR1 and FPR2 prove the impact of ECL in peptide recognition.

C) Line 203: "... in fHN (fM1-R4) and Abeta42 (D1-D5)..." Have authors thought about if replacing the first 4 residues of fHN with the first 5 residues of Abeta42 would also show binding and signalling?

After the points 1-16 mentioned above are updated, I strongly recommend Nature Communication to accept this manuscript. This is a remarkable piece of work and should be published!

Responses to reviewers' comments

Reviewer #1 (Remarks to the Author):

This is a comprehensive cryo-EM study of FPR2 with peptide ligands including A β ₄₂, the neuroprotective peptide humanin, and two smaller peptides in addition to a FPR1 and a small peptide complex. Residues responsible for affinity and signaling for FPR2 by A β ₄₂ and by humanin, and selectivity with respect to FPR1, a GPCR with ~70% identity to FPR2 are verified by mutations and IP assays. This study follows the X-ray study by Wu et al in Nat Commun on the complex of FRP-2 last year with the peptide agonist WKYMVm. The described interactions of FPR2 with peptides of various sequences and lengths provides insight into its binding promiscuity relative FPR1 and, significantly, could provide an understanding the AD pathology with the structure of A β ₄₂ and the foundation for structure-based drug design. All five structures were determined with the Gai β 1 γ 2. Since there is an allosteric relationship between the orthosteric and G-protein binding sites, the authors should discuss if there are any differences in the interactions between FPRs-Gai for all five ligands and, if there are, to describe them. They comment that helices VI and VII move to accommodate Gai, and the structures of FPR and G-proteins align well, but there are no comment/discussion of side chain interactions for each agonist. For A β ₄₂, it is internalized and interactions with arrestin might be more relevant for AD, but the absence of any discussion with Gai for all ligands in FPRs is a missed opportunity. At the very least, they should state they do not observe any differences in the interactions (and distances) among the side chains of Gai and receptors, if this is the case.

— We thank the reviewer for these comments. As suggested, we compared the interactions among the side chains of the receptor and G_i in the five structures and found that most of the interactions are conserved (see figure below). This has been added to the manuscript as “Alignment of the active FPR structures reveals an overlap of the backbones for both the receptor and G protein (Supplementary Fig. 4b), and the receptor-G_i interactions are mostly conserved in these structures, suggesting a common activation mode of the two FPRs” (lines 88-90).

Interactions between the receptor and G α in the G α _i-bound FPR1 and FPR2 structures. The structures of fMLF-FPR1-G α _i, fM5-FPR2-G α _i, fM9-FPR2-G α _i, fHN-FPR2-G α _i and A β ₄₂-FPR2-G α _i are shown in cartoon representation. Polar interactions are shown as red dashed lines. **a**, Interactions between the receptor and the C terminus of α 5-helix in G α . **b**, Hydrophobic interactions between the receptor and α 5-helix. **c**, Interactions between the receptor ICL2 and G α .

Minor corrections:

Line 60, “stays unknown” should be “is unknown”

— This has been corrected as suggested (line 55).

Line 93, “Structural alignment of the active FPR structures...” should be “Alignment of the active FPR structures...”

— This has been corrected as suggested (line 88).

Line 200, although implied by the last sentence on this line, the authors should make a statement to note the residues of humanin peptide defined by density (1-15) have not been tested biologically.

— We followed this suggestion and have added the statement “Nevertheless, the N-terminal segment M1-E15 of HN has not been tested biologically to exclude the requirement of the rest of the peptide for its functionality” to the revised version (lines 198-199).

Line 220, “the” should precede “majority”

— This has been corrected as suggested (line 219).

Line 249, the authors provide a structural explanation for the 15-fold greater affinity for humanin versus A β ₄₂ and provide the percentage of the binding interface for only the extracellular loops. They should also provide the total binding interface for humanin and A β ₄₂ as they note that humanin makes “extensive contacts” compared to A β ₄₂ in the subpocket.

— The suggestion is well taken. We have added the total binding interface for fHN and A β ₄₂ to the revised version (line 252). As discussed in this section, although the total binding interface is comparable for fHN and A β ₄₂ (fHN, 1077 Å²; A β ₄₂, 1166 Å²), the stronger binding of HN is likely gained in two regions: (i) the N-terminal residue M1 in HN, which is key for receptor activation, forms more extensive interactions with the receptor than the A β ₄₂ residue D1; (ii) the receptor extracellular region contributes more to HN binding than A β ₄₂ binding. The stronger binding of HN in the extracellular region may have a beneficial effect on initial peptide recognition and/or peptide entry into the transmembrane binding cavity as discussed in the section “Promiscuous ligand recognition of FPR2”.

Line 271, I don’t know of a case where “extracellular milieu” in this context. They should change it to “extracellular loops.”

— This has been changed as suggested (line 271).

Line 276, Y257^{6.51} in FPR1 not only has a hydroxyl group (compared to F257^{6.51} in FPR2) but there is slightly different orientation that would clash with fMLF. The authors should make a note of this.

— We followed this suggestion and have changed the statement to “Owing to a spatial hindrance caused by the larger side chains of residues F81^{2.60} and Y257^{6.51} in FPR1 (L81^{2.60} and F257^{6.51} in FPR2) and a different orientation of Y257^{6.51}, fMLF moves towards helices IV and V, excluding its contact with helix VII (Fig. 4d)” (lines 276-278).

Line 344, “promise” to be “premise”

— This has been changed as suggested (line 344).

Line 428 and 438, please explain what is meant by “technical” triplicate. Do you mean “at least in triplicate?” In addition for line 438, why was A β ₄₂ performed with n=18 and the others with an n=3?

— Thank the reviewer for this comment. We meant to use “technical triplicate” to indicate that each independent experiment was performed in triplicate. To avoid confusion, we have removed

the word “technical” from legends of all related figures and tables. Regarding the large number of independent experiments for the binding assay of A β ₄₂, we performed this assay in parallel with some FPR2 mutants to test the effect of some key residues on A β ₄₂ binding. However, the cell surface expression of these mutants was low and we didn’t obtain any robust data. Thus, we didn’t include these data in the manuscript. But all the A β ₄₂ data were included to reflect the consistency of the result.

Figure 3, The superposition in (a) is not useful. It does not show the β -strand vs the coiled structure for the peptides. I recommend another two separate figures shown the cartoon form. These two cartoon forms could be here or somewhere in Fig. 1.

— We thank the reviewer for this suggestion. Actually, the different foldings of A β ₄₂ and fHN are clearly shown in Fig. 2a and Fig. 3e. These figure are now cited in the first sentence of paragraph 2, page 10: “Unlike A β ₄₂ that recognizes FPR2 by adopting a β in-sheet structure (Fig. 2a), the majority of N-terminal part in fHN (residues R4-E15) exhibits a coiled conformation when bound to the receptor (Fig. 3e)”.

Line 536, there should “the” between “in” and “presence”

— This has been corrected as suggested (line 412).

Reviewer #2 (Remarks to the Author):

The manuscript from Zhu et al shows one cryo-EM structure of the formyl peptide receptor 1 (FPR1) and 4 structures of FPR2 with different peptide agonists bound including the beta amyloid peptide Abeta42 that forms plaque in the brain of Alzheimer’s patients. The manuscript has provided detailed insights in the ligand binding and recognition in FPRs. The authors also discovered the difference between FPR1 and FPR2 that resides at the extracellular region, where FPR1 shows a narrower entry for ligands than FPR2. Besides the cryo-EM structures, the authors provide thorough and comprehensive mutational studies to identify the impact of the residues that are involved in binding and also the signalling activity. Overall, this is an outstanding manuscript.

— We are grateful to the reviewer for the positive assessment.

As the authors are linking FPR2 with Alzheimer’s disease by solving the structure of FPR2-Gi-Abeta42, I only have a few points around Abeta40/42 that may require a bit more study and/or citation.

1) Line 137-139: “The other parts of Abeta42 may play a role in stabilizing the conformation of the peptide N terminus to enable FPR2 binding and/or is key for initial receptor-peptide recognition.” I suggest the authors to mention the folding of Abeta42, as Abeta42 can fold into a variety of structures, and apparently only the specific form of Abeta42 can activate/bind FPR2.

— The suggestion is well taken. The requirement of the specific form of A β ₄₂ for FPR2 binding/activation has been mentioned in the revised version as “The other parts of A β ₄₂ may play a role in stabilizing the conformation of the peptide N terminus to enable FPR2 binding and/or is key for initial receptor-peptide recognition. While previous structural studies of A β revealed multiple conformations of the peptides¹⁵, the A β ₄₂–FPR2–G₁₂ structure indicates that a specific form of A β ₄₂ is required for binding to FPR2” (lines 135-139).

2) The authors show the binding of Abeta42 and Abeta40 at more or less the same level (Fig. 2b). In the solved structure, Abeta42 residues 1-10 and 37-42 form an antiparallel beta-sheet. What about Abeta40? Would Abeta40 also form such an antiparallel beta-sheet and activate FPR2 as Abeta42? Does Abeta40 also work as agonist or partial agonist or antagonist/inverse agonist? This question is raised because Abeta42 is more toxic than Abeta40 in Alzheimer's disease. Therefore, it would be great to address if FPR2 can differentiate Abeta40 and Abeta42 in signalling. As the reviewer, it is unreasonable to ask the authors to solve one more structure of FPR2-Gi-Abeta40 (if it works), but it would be good to show the signalling activity of FPR2-Abeta40 in comparison with FPR2-Abeta42 by IP accumulation study as in Extended Figure 5.

— We followed the reviewer's suggestion by performing the A β ₄₀-induced IP accumulation assay of FPR2. The result shows that A β ₄₀ has an only mildly reduced agonist potency compared to A β ₄₂ (see figure below), reflecting a limited effect of the last two residues in A β ₄₂ on its agonistic activity. The new data have been added to the revised version as “Our ligand binding assay showed that A β ₄₂ and A β ₄₀, another major form of A β with two residues shorter at the C terminus, bound to FPR2 with comparable binding affinities, while the N-terminal fragment A β ₁₋₁₂ was incapable of binding to the receptor (Fig. 2b). In addition, A β ₄₀ displays an only mild reduction of potency in inducing FPR2 signalling compared to A β ₄₂ (3-fold reduction of EC₅₀; Supplementary Fig. 5i and Supplementary Table 3). These data suggest that the last two amino acids of A β ₄₂ do not contribute much to its binding and agonistic activity at FPR2, but the N-terminal region itself is not enough for recognizing the receptor” (lines 128-135). The new data are now included in Supplementary Fig. 5i and Supplementary Table 3.

A β -induced IP accumulation assay of wild-type FPR2. Dose-response curves are generated from at least four independent experiments performed in triplicate. Data are shown as mean \pm SEM. Supplementary Table 3 provides detailed numbers of independent experiments (n) and statistical evaluation. A₄₂, EC₅₀ = 31 μ M; A₄₀, EC₅₀ = 93 μ M.

Below are minor points:

3) Line 135-137: “These data suggest that the very C terminus of Abeta42 does not contribute much to its binding affinity to FPR2, ...”. Maybe change the “very C terminus” to “the last two amino acids”, as it sort of leads the reader to look for Abeta37-42 binding assay in Fig. 2b.

— The “very C terminus” has been changed to “last two amino acids” as suggested (line 133).

4) Line 184: “corelate”. Please change to “correlate”.

— The typo has been corrected (line 183).

5) Line 492-3: *Is the HA signal peptide at the C-terminus? From the written text, it looks like HA is after the C-terminus of FPR2.*

— We are sorry for the mistake. The HA signal peptide is actually at the N terminus. The sentence has been revised to “The human FPR2 gene was cloned into a modified pFastBac1 vector with an HA signal peptide at the N terminus and a PreScission protease site followed by a Flag tag and a 2×strep-tag at the C terminus” (lines 363-365).

6) Line 504-5/530-1: *Please describe the ratio between the cell pellet (gram) and suspension buffer (mL).*

— We didn’t measure the weight of cell pellet, but we kept the ratio between the volumes of cell culture and suspension buffer (10:1) consistent in all experiments. This information has been added to the Methods:

“The cell pellets from 500 ml of cell culture were thawed and suspended in 50 ml of suspension buffer containing 25 mM HEPES, pH 7.5, 150 mM NaCl, 10 mM MgCl₂, 10% (v/v) glycerol and EDTA-free protease inhibitor cocktail tablets (Roche)” in the section “Purification of fMLF–FPR1–G_{i1} complex” (lines 375-377);

“The cell pellets from 200 ml of cell culture that expresses the FPR2–G_{i2} complex were thawed on ice and suspended in 20 ml of lysis buffer containing 20 mM HEPES, pH 7.5, 50 mM NaCl, 2 mM MgCl₂, 25 mU ml⁻¹ apyrase, TEV protease (custom-made) and EDTA-free protease inhibitor cocktail tablets, and incubated at room temperature for 1 h” in the section “Purification of peptide agonist–FPR2–G_{i2} complexes” (lines 404-407).

7) Line 508/534-5: *Please describe the ratio between membrane (gram or mg/mL) and resuspension buffer.*

— We didn’t measure the weight of membrane, but we kept the ratio between the volumes of cell culture and resuspension buffer (10:1) consistent in all experiments. This information has been added to the Methods:

“The membranes were resuspended and solubilized in 50 ml of solubilization buffer containing 25 mM HEPES, pH 7.5, 150 mM NaCl, 10 mM MgCl₂, 10% (v/v) glycerol, 25 mU ml⁻¹ apyrase, 0.5% (w/v) *n*-dodecyl-β-D-maltopyranoside (DDM) (Anatrace), 0.1% (w/v) cholesterol hemisuccinate (CHS) (Sigma) and 50 μM fMLF at 4 °C for 2 h” in the section “Purification of fMLF–FPR1–G_{i1} complex” (lines 380-384);

“The cell membranes were collected by centrifugation at 160,000g for 30 min, and resuspended in 20 ml of resuspension buffer containing 50 mM HEPES, pH 7.5, 150 mM NaCl, 10 mM MgCl₂, 25 mU ml⁻¹ apyrase and EDTA-free protease inhibitor cocktail” in the section “Purification of peptide agonist–FPR2–G_{i2} complexes” (lines 408-411).

8) Line 512/539: *“The supernatant was...”: Please change “supernatant” to “solubilized fraction”.*

— This has been changed as suggested (lines 384 and 414).

9) Line 513/540: *Please describe how much resin is used for the material from __ gram of cell pellet.*

— As above, we kept the ratio between the volumes of cell culture and strep resin (500:1) consistent in all experiments. The amounts of resin (1 ml for the FPR1 complex and 400 μ l for the FPR2 complexes) have been added to the Methods (lines 385 and 415).

10) *Line 521: Is it 200 mM Tris-HCl or 20 mM Tris-HCl?*

— We have confirmed that high concentration of Tris-HCl (200 mM) was used to ensure complete dissolution of biotin.

11) *Line 522/550: Between Strep-tag purification and SEC, is there any concentration step introduced?*

— Yes, the protein samples were concentrated to 500 μ l using a 100-kDa molecular weight cut-off concentrator (Millipore) before subjected to SEC. This information has been added to the Methods (lines 395 and 421).

12) *Line 526: Please describe the concentrator used.*

— The concentrator (100-kDa molecular weight cut-off concentrator (Millipore)) has been described as suggested (line 399).

13) *Line 527: Please describe the column used for aSEC, here or in Extended Figure 1 legend.*

— The column used for aSEC (4.6 \times 250 mm Nanofilm SEC-250 column (Sepax Technologies)) has been described in the Methods (lines 402 and 427) and the legend of Supplementary Fig. 1.

14) *Line 533: using a “Dounce homogenizer”.*

— This has been corrected as suggested (line 408).

15) *Line 562: Please add “Angstrom” to the pixel size.*

— The “Å” has been added (line 439).

16) *Extended Data Fig. 1a/b/c: Please describe ligands used here.*

— The ligand fMLF was used for FPR1 purification and fHN was used for FPR2 purification. This information has been added to the legend of Supplementary Fig. 1a-c.

Suggestions for the future work:

A) *The authors show that Abeta42 residues 1-10 and 37-42 form an antiparallel beta-sheet (Fig. 2a), and also Abeta12 alone is not enough to bind FPR2 (Fig. 2b). This is a very important and remarkable finding. Have the authors tested mixing the N-terminal peptide Abeta1-12 and the C-terminal Abeta37-42 and see if such a mixture can activate FPR2? If N-terminal or C-terminal peptides alone are not enough to activate FPR2 but requires the synergy from both terminal peptides, then it will demonstrate the distinct function of Abeta42 from other Abeta peptides in causing AD.*

B) *Line 321-322: “..., demonstrating the importance of the receptor extracellular region in recognizing the long peptides.” It would be great to see if swapping the ECLs between FPR1 and FPR2 prove the impact of ECL in peptide recognition.*

C) *Line 203: “... in fHN (fM1-R4) and Abeta42 (D1-D5)...” Have authors thought about if replacing the first 4 residues of fHN with the first 5 residues of Abeta42 would also show binding and signalling?*

— We thank the reviewer for these suggestions. We have ordered synthesis of A β ₃₇₋₄₂ and the N-terminally substituted fHN, and are working on cloning of the ECL-chimeric receptors. The functional assays are also in progress.

After the points 1-16 mentioned above are updated, I strongly recommend Nature Communication to accept this manuscript. This is a remarkable piece of work and should be published!

— We thank the reviewer for the positive assessment.

REVIEWERS' COMMENTS

Reviewer #1 (Remarks to the Author):

This revision has responded to the reviewers' comments in the original submission. The work is outstanding.

A minor error was detected in Fig. 4 during the review of the revision. In the figure legend and in figure 4c, the fM9-L7 residue is mentioned as I7.

Reviewer #2 (Remarks to the Author):

I thank all the authors for the input and correction, as well as the new data. The revised version has satisfied all the points raised up in my previous feedback. I suggest Nature Communication to accept the revised manuscript. Wish you all the best.

Response to reviewers' comments

Reviewer #1 (Remarks to the Author):

This revision has responded to the reviewers' comments in the original submission. The work is outstanding.

— We are grateful to the reviewer for the positive assessment.

A minor error was detected in Fig. 4 during the review of the revision. In the figure legend and in figure 4c, the fM9-L7 residue is mentioned as I7.

— The typo has been corrected in both the figure and legend.

Reviewer #2 (Remarks to the Author):

I thank all the authors for the input and correction, as well as the new data. The revised version has satisfied all the points raised up in my previous feedback. I suggest Nature Communication to accept the revised manuscript. Wish you all the best.

— We are grateful to the reviewer for the positive assessment.